# DIPS-Plus: The Enhanced Database of Interacting Protein Structures for Interface Prediction

**Alex Morehead**[*]
University of Missouri
acmwhb@missouri.edu

**Chen Chen**
University of Missouri
chen.chen@umsystem.edu

**Ada Sedova**
Oak Ridge National Laboratory
sedovaaa@ornl.gov

**Jianlin Cheng**
University of Missouri
chengji@missouri.edu

## Abstract

How and where proteins interface with one another can ultimately impact the proteins' functions along with a range of other biological processes. As such, precise computational methods for protein interface prediction (PIP) come highly sought after as they could yield significant advances in drug discovery and design as well as protein function analysis. However, the traditional benchmark dataset for this task, Docking Benchmark 5 (DB5) [1], contains only a modest 230 complexes for training, validating, and testing different machine learning algorithms. In this work, we expand on a dataset recently introduced for this task, the Database of Interacting Protein Structures (DIPS) [2, 3], to present DIPS-Plus, an enhanced, feature-rich dataset of 42,112 complexes for geometric deep learning of protein interfaces. The previous version of DIPS contains only the Cartesian coordinates and types of the atoms comprising a given protein complex, whereas DIPS-Plus now includes a plethora of new residue-level features including protrusion indices, half-sphere amino acid compositions, and new profile hidden Markov model (HMM)-based sequence features for each amino acid, giving researchers a large, well-curated feature bank for training protein interface prediction methods. We demonstrate through rigorous benchmarks that training an existing state-of-the-art (SOTA) model for PIP on DIPS-Plus yields SOTA results, surpassing the performance of all other models trained on residue-level and atom-level encodings of protein complexes to date.

## 1 Introduction

Proteins are one of the fundamental drivers of work in living organisms. Their structures often reflect and directly influence their functions in molecular processes, so understanding the relationship between protein structure and protein function is of utmost importance to biologists and other life scientists. Here, we study the interaction between binary protein complexes, pairs of protein structures that bind together, to better understand how these coupled proteins will function *in vivo*. Predicting where two proteins will interface *in silico* has become an appealing method for measuring the interactions between proteins as a computational approach saves time, energy, and resources compared to traditional methods for experimentally measuring such interfaces [4].

A key motivation for determining protein-protein interface regions is to decrease the time required to discover new drugs and to advance the study of newly designed and engineered proteins [5].

---

[*]https://amorehead.github.io/

Submitted to the 35th Conference on Neural Information Processing Systems (NeurIPS 2021) Track on Datasets and Benchmarks. Do not distribute.

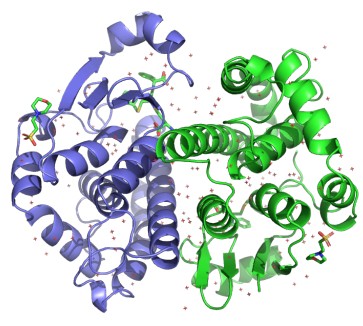

Figure 1: A PyMOL [6] visualization for a complex of interacting proteins (PDB ID: 10GS).

Towards this end, we set out to curate a dataset large enough and with enough features to develop a computational model that can reliably predict the residues that will form the interface between two given proteins. In response to the exponential rate of progress being made in applying representation learning to biomedical data, we designed a dataset to accommodate the need for more detailed features indicative of interacting protein residues to solve this fundamental problem in structural biology.

## 2 Related Work

Machine learning has been used heavily to study biomolecules such as DNA, RNA, proteins, and drug-like bio-targets. From a classical perspective, a wide array of machine learning algorithms have been employed in this domain. [7, 8] used Bayesian networks to model gene expression data. [9] give an overview of HMMs being used for biological sequence analysis, such as in [10]. [11] have used decision trees to classify membrane proteins. In a similar vein, Liu *et al.* [12] used support vector machines (SVMs) to automate the recognition of protein folds.

In particular, machine learning methods have also been used extensively to help facilitate a biological understanding of protein-protein interfaces. [13] created a random forests model for interface region prediction using structure-based features. Chen *et al.* [14] trained SVMs solely on sequence-based information to predict interfacing residues. Using both sequence and structure-based information, [15] created an SVM for partner-specific interface prediction. Shortly after, [16] achieved even better results by adopting an XGBoost algorithm and classifying residue pairs structured as pairs of feature vectors.

Another avenue of research related to interface prediction stems from traditional computational approaches to protein docking. Such domain methods have previously been used to achieve global docking results between two or more protein structures, and interface predictors have found great use within such docking software. However, the performance of interface predictors remains a notable shortcoming of these traditional docking methods [1, 17]. Hence, innovations in interface prediction via new machine learning methods and enhanced protein complex datasets on which they are trained could lead to improved performance of future docking software.

Over the past several years, deep learning has established itself as an effective means of automatically learning useful feature representations from data, with the MSA Transformer presenting a prime example of successful unsupervised learning on protein sequences [18]. Rivaling classical features, these learned feature representations, which oftentimes describe complex interactions and relationships between entities, can be used for a range of tasks including classification, regression, generative modeling, and even advanced tasks such as playing Go [19] or folding proteins *in silico* [20]. On the other hand, unsupervised representation learning can facilitate SOTA supervised prediction of mutational effect and secondary structure, as well as long-range contact prediction [21]. Thus, creating a dataset that provides sufficient information regarding complex prediction for unsupervised or semi-supervised learning is also important to the supervised learning task, since the combination

of information-rich features and graph-based protein structural data makes large-scale training on generative graph models possible.

Out of all the promising domains of deep learning, one area in particular, geometric deep learning, has arisen as a natural avenue for modeling scientific among other types of relational data [22], such as the protein complex shown in Figure 1. Previously, geometric learning algorithms like convolutional neural networks (CNNs) and graph neural networks (GNNs) have been used to predict protein interfaces. Fout *et al.* [23] designed a siamese GNN architecture to learn weight-tied feature representations of residue pairs. This approach processes subgraphs for the residues in each complex and aggregates node-level features locally using a nearest-neighbors approach. Since this partner-specific method derives its training dataset from DB5, it is ultimately data-limited. [2] represent interacting protein complexes by voxelizing each residue into a 3D grid and encoding in each grid entry the presence and type of the residue's underlying atoms. This partner-specific encoding scheme captures structural features of interacting complexes, but it is not able to scale well due to its requiring a computationally-expensive spatial resolution of the residue voxels to achieve good results.

Continuing the trend of applying geometric learning to protein structures, [24] perform partner-independent interface region prediction with an attention-based GNN. This method learns to perform binary classification of the residues in both complex structures to identify regions where residues from both complexes are likely to interact with one another. However, because this approach predicts partner-independent interface regions, it is less likely to be useful in helping solve related tasks such as drug-protein interaction prediction and protein-protein docking [25]. To date, the best results obtained by any model for protein interface prediction come from [26] where high-order (i.e. sequential and coevolution-based) interactions between residues are learned and preserved throughout the network in addition to structural features embedded in protein complexes. However, this approach is also data-limited as it uses the DB5 dataset to derive its training data. As such, it remains to be shown how much precision could be obtained with these and similar methods by training them on much more exhaustive datasets.

## 3 Dataset

### 3.1 Overview

As we have seen, two main encoding schemes have been proposed for protein interface prediction: modeling protein structures at the atomic level and modeling structures at the level of the residue. Modeling protein structures in terms of their atoms can yield a detailed representation of such geometries, however, accounting for each atom in a structure can quickly become computationally burdensome or infeasible for large structures. On the other hand, as residues are comprised of multiple atoms, modeling only a structure's residues allows one to employ their models on a more computationally succinct view of the structure, thereby reducing memory requirements for the training and inference of biomolecular machine learning models by focusing only on the alpha-carbon (CA) atoms of each residue. The latter scheme also enables researchers to curate robust residue-based features for a particular task, a notion of flexibility quite important to the success of prior works in protein bioinformatics [15, 23, 26, 27]. Nonetheless, both schemes, when adopted by a machine learning algorithm such as a neural network, require copious amounts of training examples to generalize past the training dataset. However, only a handful of extensive datasets for protein interface prediction currently exist, DIPS being the largest of such examples, and it is designed solely for modeling structures at the atomic level. If one would like to model complexes at the residue level to summarize the structural and functional properties of each residue's atoms as additional features for training, DB5 is currently one of the only datasets with readily-available pairwise residue labels that meets this criterion. As such, one of the primary motivations for curating DIPS-Plus was to answer the following two questions: Why must one choose between having the largest possible dataset and having enough features for their interface prediction models to generalize well? And is it possible for a single dataset to facilitate both protein-encoding schemes while maintaining its size and feature-richness?

### 3.2 Usage

As a follow-up to the above two questions, we constructed DIPS-Plus, a feature-expanded version of DIPS accompanied, with permission from the original authors of DIPS, by a CC-BY 4.0 license

Table 1: Residue features added in DIPS-Plus

| New Features (1) | New Features (2) |
|---|---|
| Secondary Structure | Half-Sphere Amino Acid Composition |
| Relative Solvent Accessibility | Coordinate Number |
| Residue Depth | Profile HMM Features |
| Protrusion Index | Amide Normal Vector |

for reproducibility and extensibility. This dataset can be used with most deep learning algorithms, especially geometric learning algorithms (e.g. CNNs, GNNs), for studying protein structures, complexes, and their inter/intra-protein interactions at scale. It can also be used to test the performance of new or existing geometric learning algorithms for node classification, link prediction, object recognition, or similar benchmarking tasks. The standardized task for which DIPS-Plus is designed is dense prediction of all possible interactions between inter-protein residues (e.g. $M \times N$ possible interactions where $M$ and $N$ are the numbers of residues in a complex's first and second structure, respectively) [26]. In the context of computer vision, then, DIPS-Plus can be seen as a dataset for pixel-wise prediction on 2D biological images. The primary metric used to score DIPS-Plus algorithms is the median area under the receiver operating characteristic curve (MedAUROC) to prevent test results for extraordinarily large complexes from having a disproportionate effect on the algorithm's overall test MedAUROC [15, 23, 2, 26]. To facilitate convenient training of future methods trained on DIPS-Plus, we provide a standardized 80%-20% cross-validation split of the DIPS-Plus complexes' file names. For these splits, we *a priori* filter out 663 complexes containing more than 1,000 residues to mirror DB5 in establishing an upper bound on the computational complexity of algorithms trained on the dataset. As is standard for interface prediction [15, 23, 2, 26], we define the labels in DIPS-Plus to be the IDs (i.e. Pandas DataFrame row IDs [28]) of inter-protein residue pairs that, in the complex's bound state, can be found within 6 Å of one another, using each residue's non-hydrogen atoms for performing distance measurements (since hydrogen atoms are often not present in experimentally-determined structures).

Similar to [2], in the version of DB5 we update with new features from DIPS-Plus (i.e. DB5-Plus), we record the file names of the complexes added between versions 4 and 5 of Docking Benchmark as the final test dataset for users' convenience. The rationale behind this choice of test dataset is given by the following points: (1) The task of interface prediction is to predict how two *unbound* (i.e. not necessarily conformal) proteins will bind together by predicting which pairs of residues from each complex will interact with one another upon binding; (2) DIPS-Plus consists solely of *bound* protein complexes (i.e. those already conformed to one another), so we must test on a dataset consisting of *unbound* complexes after training to verify the effectiveness of the method for PIP; (3) Each of DB5-Plus' *unbound* test complexes are of varying interaction types and difficulties for prediction (e.g. antibody-antigen, enzyme substrate), simulating how future unseen proteins (i.e. those in the wild) might be presented to the model following its training; (4) DB5's test complexes (i.e. those added between DB4 and DB5) represent a time-based data split also used for evaluation in [23, 2, 26], so for fair comparison with previous SOTA methods we chose the same complexes for testing.

### 3.3 Construction

In total, DIPS-Plus consists of 42,112 complexes compared to the 42,826 complexes in DIPS after pruning out 714 large and evolutionarily-distinct complexes that are no longer available in the RCSB PDB (as of April 2021) or for which multiple sequence alignment (MSA) generation was prohibitively time-consuming and computationally expensive. The original DIPS, being a carefully curated PDB subset, contains almost 200x more protein complexes than the modest 230 complexes in DB5, what is still considered to be a gold standard of protein-protein interaction datasets. Other protein binding datasets such as PDBBind [29] (containing 5,341 protein-protein complexes) and that which was used in the development of MaSIF [30] (containing roughly 12,000 protein-protein complexes in total) have previously been curated for machine learning of protein complexes. However, to the best of our knowledge, DIPS-Plus serves as the single largest database of PDB protein-protein complexes incorporating novel features such as profile HMM-derived sequence conservation and half-sphere amino acid compositions shown to be indicative of residue-residue interactions in Section 4. It is still a possibility that PDBBind or MaSIF may contain useful information regarding complexes not

already contained in DIPS-Plus. Fortunately, it remains possible with our data pipeline to extend DIPS-Plus to include these new complexes in PDBBind or MaSIF. For the time being, we defer the exploration of this idea to future works.

## 3.4 Quality

Regarding the quality of the complexes in DIPS-Plus, we employ a similar pruning methodology as [2] to ensure data integrity. DIPS-Plus, along with the works of others [1, 29, 30], derives its complexes from the PDB which conducts statistical quality summaries in its structure deposition processes and post-deposition analyses [31]. Nonetheless, recent studies on the PDB have discovered that the quality of its structures can, in some cases, vary considerably between structures [32]. As such, in selecting complexes to include in DIPS-Plus, we perform extensive filtering after obtaining the initial batch of 180,000 complexes available in the PDB. Such filtering includes (1) removing PDB complexes containing a protein chain with more than 30% sequence identity with any protein chain in DB5-Plus per [33, 34], (2) selecting complexes with an X-ray crystallography or cryo-electron microscopy resolution greater than 3.5 Å (i.e. a standard threshold in the field), (3) choosing complexes containing protein chains with more than 50 amino acids (i.e. residues), (4) electing for complexes with at least 500 Å$^2$ of buried surface area, and (5) picking only the first model for a given complex. The motivation for the first filtering step is to ensure that we do not allow training datasets built from DIPS-Plus to bias the DB5-Plus test results of models trained on DIPS-Plus, with the remaining steps carried out to follow conventions in the field of protein bioinformatics.

## 3.5 New Features

The features we chose to add to DIPS to create DIPS-Plus were selected carefully and intentionally based on our analysis of previously-successful interface prediction models. In this section, we describe each of these new features in detail, including why we chose to include them, how we collected or generated them, and the strategy we took for normalizing the features and imputing missing feature values when they arose. These features were derived only for standard residues (e.g. amino acids) by filtering out hetero residues and waters from each PDB complex before calculating, for example, half-sphere amino acid compositions for each residue. This is, in part, to reduce the computational overhead of generating each residue's features. More importantly, however, we chose to ignore hetero residue features in DIPS-Plus to keep it consistent with DB5 as hetero residues and waters are not present in DB5.

DIPS-Plus, compared to DIPS, not only contains the original Protein Data Bank (PDB) features in DIPS such as amino acids' Cartesian coordinates and their corresponding atoms' element types but now also new residue-level features shown in Table 1 following a feature set similar to [15, 23, 26]. DIPS-Plus also replaces the residue sequence-conservation feature conventionally used for interface prediction with a novel set of emission and transition probabilities derived from HMM sequence profiles. Each HMM profile used to ascertain these residue-specific transition and emission probabilities are constructed by HHmake [35] using MSAs that were generated after two iterations by HHblits [35] and the Big Fantastic Database (BFD) (version: March 2019) of protein sequences [36]. Inspired by the work of Guo *et al.* [27], we chose to use HMM profiles to create sequence-based features in DIPS-Plus as they have been shown to contain more detailed information concerning the relative frequency of each amino acid in alignment with other protein sequences compared to what has traditionally been done to generate sequence-based features for interface prediction, directly sampling (i.e. windowing) MSAs to assess how conserved (i.e. buried) each residue is [35].

### 3.5.1 Secondary Structure

Secondary structure (SS) is included in DIPS-Plus as a categorical variable that describes the type of local, three-dimensional structural segment in which a residue can be found. This feature has been shown to correlate with the presence or absence of protein-protein interfaces [37]. In addition, the secondary structures of residues have been found to be informative of the physical interactions between main-chain and side-chain groups [38]. This is one of the primary motivations for including them as a residue feature in DIPS-Plus. As such, we hypothesize adding secondary structure as a feature for interface prediction models could prove beneficial to model performance as it would allow them to more readily discover interactions between structures' main-chain and side-chain groups.

We generate each residue's SS value using version 3.0.0 of the Database of Secondary Structure Assignments for Proteins (DSSP) [39], a well-known and frequently-used software package in the bioinformatics community. In particular, we use version 1.78 of BioPython [40] to call DSSP and have it retrieve for us DSSP's results for each residue. Each residue is assigned one of eight possible SS values, 'H', 'B', 'E', 'G', 'I', 'T', 'S', or '-', with the symbol '-' signifying the default value for unknown or missing SS values. Since this categorical feature is naturally one-hot encoded, it does not need to be normalized numerically.

### 3.5.2 Relative Solvent Accessibility

Each residue can behave differently when interacting with water. Solvent accessibility is a scalar (i.e. type 0) feature that quantifies a residue's accessible surface area, the area of a residue's atoms that can be touched by water. Polar residues typically have larger accessible surface areas, while hydrophobic residues tend to have a smaller accessible surface area. It has been observed that hydrophobic residues tend to appear in protein interfaces more often than polar residues [41]. Including solvent accessibility as a residue-level feature, then, may provide models with additional information regarding how likely a residue is to interact with another inter-protein residue.

Relative solvent accessibility (RSA) is a simple modification of solvent accessibility that normalizes each residue's solvent accessibility by an experimentally-determined normalization constant specific to each residue. These normalization constants are designed to help more closely correlate generated RSA values with their residues' true solvent accessibility [42]. Here, we again use BioPython and DSSP together, this time to generate each residue's RSA value. The RSA values returned from BioPython are pre-normalized according to the constants described in [42] and capped to an upper limit of 1.0. Missing RSA values are denoted by the NaN constant from NumPy [43], a popular scientific computing library for Python. As we use NumPy's representation of NaN for missing values, users have available to them many convenient methods for imputing missing feature values for each feature type, and we provide scripts with default parameters to do so with our source code for DIPS-Plus. By default, NaN values for numeric features like RSA are imputed using the feature's columnwise median value.

### 3.5.3 Residue Depth

Residue depth (RD) is a scalar measure of the average distance of the atoms of a residue from its solvent-accessible surface. Afsar *et al.* [15] have found that for interface prediction this feature is complementary to each residues' RSA value. Hence, this feature holds predictive value for determining interacting protein residues as it can be viewed as a description of how "buried" each residue is. We use BioPython and version 2.6.1 of MSMS [44] to generate each residue's depth, where the default quantity for a missing RD value is NaN. To make all RD values fall within the range [0, 1], we then perform structure-specific min-max normalization of each structure's non-NaN RD values using scikit-learn [45]. That is, for each structure, where $min = 0$ and $max = 1$, we find its minimum and maximum RD values and normalize the structure's RD values $X$ using the expression

$$X = \frac{X - X.min(axis = 0)}{X.max(axis = 0) - X.min(axis = 0)} \times (max - min) + min.$$

### 3.5.4 Protrusion Index

A residue's protrusion index (PI) is defined using its non-hydrogen atoms. It is a measure of the proportion of a 10 Å sphere centered around the residue's non-hydrogen atoms that is not occupied by any atoms. By computing residues' protrusion this way, we end up with a 1 x 6 feature vector that describes the following six properties of a residue's protrusion: average and standard deviation of protrusion, minimum and maximum protrusion, and average and standard deviation of the protrusion of the residue's non-hydrogen atoms facing its side chain. We used version 1.0 of PSAIA [46] to calculate the PIs for each structure's residues collectively. That is, each structure has its residues' PSAIA values packaged in a single .tbl file. Missing PIs default to a 1 x 6 vector consisting entirely of NaNs. We min-max normalize each PI entry columnwise to get six updated PI values, similar to how we normalize RD values in a structure-specific manner.

### 3.5.5 Half-Sphere Amino Acid Composition

Half-sphere amino acid compositions (HSAACs) are comprised of two 1 x 21 unit-normalized vectors concatenated together to get a single 1 x 42 feature vector for each residue. The first vector, termed the upward composition (UC), reflects the number of times a particular amino acid appears along the residue's side chain, while the second, the downward composition (DC), describes the same measurement in the opposite direction, with the 21st vector entry for each residue corresponding to the unknown or unmappable residue, '-'. Knowing the composition of amino acids along and away from a residue's side chain, for all residues in a structure, is another feature that has been shown to offer crucial predictive value to machine learning models for interface prediction as it can describe physiochemical and geometric patterns in such regions [47]. These UC and DC vectors can also vary widely for residues, suggesting an alternative way of assessing residue accessibility [15, 26]. Missing HSAACs default to a 1 x 42 vector consisting entirely of NaNs. Furthermore, since both the UC and DC vectors for each residue are unit normalized before concatenating them together, after concatenation all columnwise HSAAC values for a structure still inclusively fall between 0 and 1.

### 3.5.6 Coordinate Number

A residue's coordinate number (CN) is conveniently determined alongside the calculation of its HSAAC. It denotes how many other residues to which the given residue was found to be significant. Significance, in this context, is defined in the same way as [15]. That is, the significance score for two residues is defined as

$$s = e^{\frac{-d^2}{2 \times st^2}},$$

where $d$ is the minimum distance between any of their atoms and $st$ is a given significance threshold which, in our case, defaults to the constant $1e^{-3}$. Then, if two residues' significance score falls above $st$, they are considered significant. As per our convention in DIPS-Plus, the default value for missing CNs is NaN, and we min-max normalize the CN for each structure's residues.

### 3.5.7 Profile HMM Features

MSAs can carry rich evolutionary information regarding how each residue in a structure is related to all other residues, and sequence profile HMMs have increasingly found use in representing MSAs' evolutionary information in a concise manner [35, 20]. In previous works on PIP, knowing the conservation of a residue has been found to be beneficial in predicting whether the residue is likely to be found in an interface [15, 23, 26], and profile HMMs capture this sequence conservation information in a novel way using MSAs. As such, to gather sequence profile features for DIPS-Plus, we derive profile HMMs for each structures' residues using HH-suite3 by first generating MSAs using HHblits followed by taking the output of HHblits to create profile HMMs using HHmake. From these profile HMMs, we can then calculate each structure's residue-wise emission and transition profiles. A residue's emission profile, represented as a 1 x 20 feature vector of probability values, illustrates how likely the residue is across its evolutionary history to emit one of the 20 possible amino acid symbols. Similarly, each residue's transition profile, a 1 x 7 probability feature vector, depicts how likely the residue is to transition into one of the seven possible HMM states.

To derive each structure's emission and transition probabilities, for a residue $i$ and a standard amino acid $k$ we extract the profile HMM entry $(i, k)$ (i.e. the corresponding frequency) and convert the frequency into a probability value with the equation

$$p_{ik} = 2^{-\frac{Freq_{ik}}{m}}.$$

where $m$ is the number of MSAs used to generate each profile HMM ($m = 1,000$ by default).

After doing so, we get a 1 x 27 vector of probability values for each residue. Similar to other features in DIPS-Plus, missing emission and transition probabilities for a single residue default to a 1 x 27 vector comprised solely of NaNs. Moreover, since each residue is assigned a probability vector as its sequence features, we do not need to normalize these sequence feature vectors columnwise. We chose to leave out three profile HMM values for each residue representing the diversity of the alignment with respect to HHmake's generation of profile HMMs from HHblits' generated MSAs for a given

Table 2: A comparison of datasets for PIP

| Dataset | # Complexes | # Residues | # Residue Interactions | # Residue Features |
|---|---|---|---|---|
| DB5 | 230 | 121,943 | 21,091 | 0 |
| DB5-Plus | 230 | 121,943 | 21,091 | 8 |
| DIPS | 42,826 | 22,547,678 | 5,767,039 | 0 |
| DIPS-Plus | 42,112 | 22,127,737 | 5,677,450 | 8 |

Table 3: How many residue features were successfully generated for each PIP dataset

| DB5-Plus | DIPS-Plus | DB5-Plus | DIPS-Plus |
|---|---|---|---|
| SS: 95,614 | SS: 17,835,959 | HSAAC: 121,943 | HSAAC: 21,791,175 |
| RSA: 121,591 | RSA: 22,104,449 | CN: 121,943 | CN: 22,127,737 |
| RD: 121,601 | RD: 22,069,320 | HMM: 121,943 | HMM: 22,127,050 |
| PI: 121,943 | PI: 19,246,789 | NV: 113,376 | NV: 20,411,267 |

structure. Since we do not see any predictive value in including these as residue features, we left them out of both DIPS-Plus and DB5-Plus.

### 3.5.8 Amide Normal Vector

Each residue's amide plane has a normal vector (NV) that we can derive by taking the cross product of the difference between the residue's CA atom and beta-carbon (CB) atoms' Cartesian coordinates and the difference between the coordinates of the residue's CB atom and its nitrogen (N) atom. If users choose to encode the complexes in DIPS-Plus as pairs of graphs, these NVs can then be used to define rich edge features such as the angle between the amide plane NVs for two residues [23]. Similar to how we impute other missing feature vectors, the default value for an underivable NV (e.g. for Glycine residues that do not have a beta-carbon atom) is a 1 x 3 vector consisting of NaNs. Further, since these vectors represent residues' amide plane NVs, we leave them unnormalized for, at users' discretion, additional postprocessing (e.g. custom normalization) of these NVs.

### 3.6 Analysis

Table 2 gives a brief summary of the datasets available for protein interface prediction to date and the number of residue features available in them. In it, we can see that our version of DIPS, labeled DIPS-Plus, contains many more residue features than its original version at the expense of minimal pruning to the number of complexes available for training. Complementary to Table 2, Table 3 shows how many features we were able to include for each residue in DB5-Plus and DIPS-Plus, respectively. Regarding DB5-Plus, we see that for relative solvent accessibility, residue depth, protrusion index, half-sphere amino acid composition, coordinate number, and profile HMM features, the majority of residues have valid (i.e. non-NaN) entries. That is, more than 99.7% of all residues in DB5-Plus have valid values for these features. In addition, secondary structures and amide plane normal vectors exist, respectively, for 78.4% and 93% of all residues. Concerning DIPS-Plus, relative solvent accessibilities, residue depths, half-sphere amino acid compositions, coordinate numbers, and profile HMM features exist for more than 98.5% of all residues. Also, we notably observe that valid secondary structures, protrusion indices, and normal vectors exist, respectively, for 80.6%, 87%, and 92.2% of all residues.

From the above analysis, we made a stand-alone observation. For both DB5-Plus and DIPS-Plus, residues' secondary structure labels are available from DSSP for, on average, 80.6% of all residues in DIPS-Plus and DB5-Plus, collectively. This implies that there may be benefits to gain from varying how we collect secondary structures for each residue, possibly by using deep learning-driven alternatives to DSSP that predict the secondary structure to which a residue belongs, as in [27]. Complementing DSSP in this manner may yield even better secondary structure values for DIPS-Plus and DB5-Plus. We defer the exploration of this idea to future work.

Table 4: The effect of our new feature set (i.e. DIPS-Plus) on a SOTA algorithm for PIP

| Method | MedAUROC |
| --- | --- |
| NGF [48] | 0.865 (0.007) |
| DTNN [49] | 0.867 (0.007) |
| Node and Edge Average [23] | 0.876 (0.005) |
| BIPSPI [16] | 0.878 (0.003) |
| SASNet* [2] | 0.885 (0.009) |
| NeiA+HOPI [26] | 0.902 (0.012) |
| NeiWA+HOPI [26] | 0.908 (0.019) |
| **NeiA+HOPI+DIPS-Plus** | **0.9473 (0.001)** |

## 4 Benchmarks

To measure the effect that DIPS-Plus has on the performance of existing machine learning methods for PIP, we trained one of the latest SOTA methods, NeiA, for 10 epochs on our standardized 80%-20% cross-validation split of DIPS-Plus' complexes to observe NeiA's behavior on DB5-Plus's test complexes thereafter. We ran this experiment three times, each with a random seed and a single GNN layer, for a fair comparison of the experiment's mean and standard deviation (i.e. in parentheses) in terms of MedAUROC. Our results from this experiment are shown in the last row of Table 4. For the experiment, we used the following architecture and hyperparameters: (1) 1 NeiA GNN layer; (2) 3 residual CNN blocks, each employing a 2D convolution module, ReLU activation function, another 2D convolution module, followed by adding the block input's identity map back to the output of the block (following a design similar to that of [26]); (3) an intermediate channel dimensionality of 212 for each residual CNN block; (4) a learning rate of 1e-5; (5) a batch size of 32; (6) a weight decay of 1e-7; and (7) a dropout (forget) probability of 0.3.

All baseline results on the DB5 test complexes in Table 4 (i.e. complexes comprised of original DB5 residue features) [48, 49, 23, 16, 26] are taken from [26], with the exception of SASNet's results from training on the original DIPS dataset. These results are denoted by an asterisk in Table 4 to indicate that they were instead taken from [2]. The best performance is in bold. In this table, we see that a simple substitution of training and validation datasets enhances the MedAUROC of NeiA when adopting its accompanying high-order pairwise interaction (HOPI) module for learning inter-protein residue-residue interactions. For reference, to the best of our knowledge, the best performance of a machine learning model trained for PIP on only the atom-level features of protein complexes is SASNet's MedAUROC of **0.885** averaged over three separate runs [2]. Such insights suggest the utility and immediate advantage of using DIPS-Plus' residue feature set for PIP over the original DIPS' atom-level feature set. Additionally, we deduce from Table 4 that the performance of previous methods for PIP is likely limited by the availability of residue-encoded complexes for training as all but one method [2] used DB5's 230 total complexes for training, validation, as well as testing.

## 5 Impact and Challenges

### 5.1 Data Representation

Over the last several years, geometric deep learning has surfaced as a powerful means of uncovering structural features in graph topologies [22]. To facilitate convenient processing of each DIPS-Plus and DB5-Plus complex to fit this and other paradigms, we include with DIPS-Plus' source code the scripts necessary to convert each complex's Pandas DataFrame into two stand-alone graph objects compatible with the Deep Graph Library (DGL) along with their corresponding residue-residue interaction matrix [50]. However, our data conversion scripts can easily be adapted to facilitate alternative data representation schemes for the complexes in DIPS-Plus and DB5-Plus. For example, one can choose to extract the graphs' node and edge features as two separate PyTorch [51] tensors for 2D or 3D convolutions, representing either the atoms or residues of each complex (i.e. user's choice) as entries in a 3D or 4D tensor, respectively. These default graph objects can then be used for a variety of graph-level tasks such as node classification (e.g. for interface region prediction) or link prediction (e.g. for inter-protein residue-residue interaction prediction). By default, each DGL graph contains for each node 86 features either one-hot encoded or extracted directly from the new feature

set described above. Further, each graph edge contains two distinct features after being min-max normalized, the angle between the amide plane NV for a given source and destination node as well as the squared relative distance between the source and destination nodes.

## 5.2 Biases

DIPS-Plus contains only bound protein complexes. On the other hand, our new PIP dataset for testing machine learning models, DB5-Plus, consists of unbound complexes. As such, the conformal state of DIPS-Plus' complexes can bias learning algorithms to learning protein structures in their final, post-deformation state since the structures in a complex often undergo deviations from their natural shape after being bound to their partner protein. However, our benchmarks in Section 4, agreeing with those of Townshend *et al.* [2], show that networks well suited to the task of learning protein interfaces (i.e. those with suitable inductive biases for the problem domain) can generalize beyond the training dataset (i.e. DIPS) and perform well on unbound protein complexes (i.e. those in DB5). Hence, through our benchmarks, we provide designers of future PIP algorithms with an example of how to make effective use of DIPS-Plus' structural bias for complexes.

## 5.3 Associated Risks

DIPS-Plus is designed to be used for machine learning of biomolecular data. It contains only publicly-available information concerning biomolecular structures and their interactions. Consequently, all data used to create DIPS-Plus does not contain any personally identifiable information or offensive content. As such, we do not foresee any negative societal impacts as a consequence of DIPS-Plus being made publicly available. Furthermore, future adaptions or enhancements of DIPS-Plus may benefit the machine learning community and, more broadly, the scientific community by providing meaningful refinements to an already-anonymized, transparent, and extensible dataset for geometric deep learning tasks in the life sciences.

## 6 Conclusion

We present DIPS-Plus, a comprehensive dataset for training and validating protein interface prediction models. Protein interface prediction is a novel, high-impact challenge in structural biology that can be vastly advanced with innovative algorithms and rich data sources. Several algorithms and even large atomic datasets for protein interface prediction have previously been proposed, however, until DIPS-Plus no single large-scale data source with rich residue features has been available. We expect the impact of DIPS-Plus to be a significantly enhanced quality of future models and community discussion in how best to design algorithmic solutions to this novel open challenge. Further, we anticipate that DIPS-Plus could be used as a template for creating new large-scale machine learning datasets tailored to the life sciences.

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

# A Appendix

## A.1 Datasheet

### A.1.1 Motivation

**For what purpose was the dataset created?** Was there a specific task in mind? Was there a specific gap that needed to be filled? Please provide a description.

DIPS-Plus was created for training and validating deep learning models aimed at predicting protein interfaces and inter-protein interactions. Without DIPS-Plus, deep learning algorithms that encode protein structures at the level of a residue would be limited either to the scarce protein complexes available in the Docking Benchmark 5 (DB5) dataset [1], to the original, feature-limited Database of Interacting Protein Structures (DIPS) dataset [2, 3], or to the smaller PDBBind or MaSIF dataset for training [29, 30].

**Who created this dataset (e.g., which team, research group) and on behalf of which entity (e.g., company, institution, organization)?**

DIPS-Plus was created by Professor Jianlin Cheng's Bioinformatics & Machine Learning (BML) lab at the University of Missouri. The original DIPS was created by Professor Ron Dror's Computational Biology lab at Stanford University and was enhanced to create DIPS-Plus with the original authors' permission.

**Who funded the creation of the dataset? If there is an associated grant, please provide the name of the grantor and the grant name and number.**

The project is partially supported by two NSF grants (DBI 1759934 and IIS 1763246), one NIH grant (GM093123), three DOE grants (DE-SC0020400, DE-AR0001213, and DE-SC0021303), and the computing allocation on the Andes compute cluster provided by Oak Ridge Leadership Computing Facility (Project ID: BIF132). In particular, this research used resources of the Oak Ridge Leadership Computing Facility at the Oak Ridge National Laboratory, which is supported by the Office of Science of the U.S. Department of Energy under Contract No. DE-AC05-00OR22725.

### A.1.2 Composition

**What do the instances that comprise the dataset represent (e.g., documents, photos, people, countries)?**
Are there multiple types of instances (e.g., movies, users, and ratings; people and interactions between them; nodes and edges)? Please provide a description.

DIPS-Plus is comprised of binary protein complexes (i.e. bound ligand and receptor protein structures) extracted from the Protein Data Bank (PDB) of the Research Collaboratory for Structural Bioinformatics (RCSB) [52]. Both protein structures in the complex are differentiable in that they are stored in their own Pandas DataFrame objects [28]. Each structure's DataFrame contains information concerning the atoms of each residue in the structure such as their Cartesian coordinates and element type. For the alpha-carbon atoms of each residue (typically the most representative atom of a residue), each structure's DataFrame also contains residue-level features like a measure of amino acid protrusion and solvent accessibility.

**How many instances are there in total (of each type, if appropriate)?**

There are 42,826 binary protein complexes in the original DIPS and 42,112 binary protein complexes in DIPS-Plus after additional pruning.

**Does the dataset contain all possible instances or is it a sample (not necessarily random) of instances from a larger set?** If the dataset is a sample, then what is the larger set? Is the sample representative of the larger set (e.g., geographic coverage)? If so, please describe how this representativeness was validated/verified. If it is not representative of the larger set, please describe why not (e.g., to cover a more diverse range of instances, because instances were withheld or unavailable).

The dataset contains all possible instances of bound protein complexes obtainable from the RCSB PDB for which it is computationally reasonable to generate residue-level features. That is, if it takes more than 48 hours to generate an RCSB complex's residue features, it is excluded from DIPS-Plus. This results in us excluding approximately 100 complexes after our pruning of RCSB complexes.

**What data does each instance consist of?** "Raw" data (e.g., unprocessed text or images)or features? In either case, please provide a description.

Each instance, consisting of a pair of Pandas DataFrames containing a series of alpha-carbon (CA) atoms and non-CA atoms with residue and atom-level features, respectively, is stored in a Python dill file for data compression and convenient file loading [53]. Each Pandas DataFrame contains a combination of numeric, categorical, and vector-like features describing each atom.

**Is there a label or target associated with each instance? If so, please provide a description.**

The dataset contains the labels of which pairs of CA atoms from opposite structures are within 6 Å of one another (i.e. positives), implying an interaction between the two residues, along with an equally-sized list of randomly-sampled non-interacting residue pairs (i.e. negatives). For example, if a complex in DIPS-Plus contains 100 interacting residue pairs (i.e. positive instances), there will also be 100 randomly-sampled non-interacting residue pairs included in the complex's dill file for optional downsampling of the negative class during training.

**Is any information missing from individual instances?** If so, please provide a description, explaining why this information is missing (e.g., because it was unavailable). This does not include intentionally removed information, but might include, e.g., redacted text.

All eight of the residue-level features added in DIPS-Plus are missing values for at least one residue. This is because not all residues have, for example, DSSP-derivable secondary structure (SS) values [39] or profile hidden Markov models (HMMs) that are derivable by HH-suite3 [35], the software package we use to generate multiple sequence alignments (MSAs) and subsequent MSA-based features. A similar situation occurs for the six other residue features. That is, not all residues have derivable features for a specific feature column, governed either by our own feature parsers or by the external feature parsers we use in making DIPS-Plus. We denote missing feature values for all features as NumPy's NaN constant with the exception of residues' SS value in which case we use '-' as the default missing feature value [43].

**Are relationships between individual instances made explicit (e.g., users' movie ratings, social network links)?** If so, please describe how these relationships are made explicit. If so, please provide a description, explaining why this information is missing (e.g., because it was unavailable). This does not include intentionally removed information, but might include, e.g., redacted text.

The relationships between individual instances (i.e. protein complexes) are made explicit by the directory and file-naming convention we adopt for DIPS-Plus. Complexes' DataFrame files are grouped into folders by shared second and third characters of their PDB identifier codes (e.g. 1x9e.pdb1_0.dill and 4x9e.pdb1_5.dill reside in the same directory x9/).

**Are there recommended data splits (e.g., training, development/validation, testing)?** If so, please provide a description of these splits, explaining the rationale behind them.

Since DIPS-Plus is relatively large (i.e. has more than 10,000 complexes), we provide a randomly-sampled 80%-20% dataset split for training and validation data, respectively, in the form of two text files: pairs-postprocessed-train.txt and pairs-postprocessed-val.txt. The file pairs-postprocessed.txt is a master list of all complex file names from which we derive pairs-postprocessed-train.txt and pairs-postprocessed-val.txt for cross-validation. It contains the file names of 42,112 complex DataFrames, filtered down from the original 42,112 complexes in DIPS-Plus to complexes having no more than 17,500 CA and non-CA atoms, to match the maximum possible number of atoms in DB5-Plus structures and to create an upper-bound on the computational complexity of learning algorithms trained on DIPS-Plus. However, we also include the scripts necessary to conveniently regenerate pairs-postprocessed.txt with a modified or removed atom-count filtering criterion and with different cross-validation ratios.

**Are there any errors, sources of noise, or redundancies in the dataset?** If so, please provide a description.

As mentioned in the missing information point above, not all residues have software-derivable features for the feature set we have chosen for DIPS-Plus. In the case of missing features, we substitute NumPy's NaN constant for the missing feature value with the exception of SS values which are replaced with the symbol '-'. We also provide with DIPS-Plus postprocessing scripts for users to perform imputation of missing feature values (e.g. replacing a column's missing values with the

column's mean, median, minimum, or maximum value or with a constant such as zero) depending on the type of the missing feature (i.e. categorical or numeric).

**Is the dataset self-contained, or does it link to or otherwise rely on external resources (e.g., websites, tweets, other datasets)?** If it links to or relies on external resources, a) are there guarantees that they will exist, and remain constant, over time; b) are there official archival versions of the complete dataset (i.e., including the external resources as they existed at the time the dataset was created); c) are there any restrictions (e.g., licenses, fees) associated with any of the external resources that might apply to a future user? Please provide descriptions of all external resources and any restrictions associated with them, as well as links or other access points, as appropriate.

The dataset relies on feature generation using external tools such as DSSP and PSAIA. However, in our Zenodo data repository for DIPS-Plus, we provide either a copy of the external features generated using these tools or the exact version of the tool with which we generated features (e.g. version 3.0.0 of DSSP for generating SS values using version 1.78 of BioPython). The most time-consuming and computationally-expensive features to generate, profile HMMs and protrusion indices, are included in our Zenodo repository for users' convenience. We also provide the final, postprocessed version of each DIPS-Plus complex in our Zenodo data bank.

**Does the dataset contain data that might be considered confidential (e.g., data that is protected by legal privilege or by doctor-patient confidentiality, data that includes the content of individuals non-public communications)? If so, please provide a description.**

No, DIPS-Plus does not contain any confidential data. All data with which DIPS-Plus was created is publicly available.

**Does the dataset contain data that, if viewed directly, might be offensive, insulting, threatening, or might otherwise cause anxiety?** If so, please describe why.

No, DIPS-Plus does not contain data that, if viewed directly, might be offensive, insulting, threatening, or might otherwise cause anxiety.

**Does the dataset relate to people?** If not, you may skip the remaining questions in this section.

No, DIPS-Plus does not contain data that relates directly to individuals.

### A.1.3 Collection Process

**How was the data associated with each instance acquired?** Was the data directly observable (e.g., raw text, movie ratings), reported by subjects (e.g., survey responses), or indirectly inferred/derived from other data (e.g., part-of-speech tags, model-based guesses for age or language)? If data was reported by subjects or indirectly inferred/derived from other data, was the data validated/verified? If so, please describe how.

The data associated with each instance was acquired from the RCSB's PDB repository for protein complexes (https://ftp.wwpdb.org/pub/pdb/data/biounit/coordinates/divided/), where each complex was screened, inspected, and analyzed by biomedical professionals and researchers before being deposited into the RCSB PDB.

**What mechanisms or procedures were used to collect the data (e.g., hardware apparatus or sensor, manual human curation, software program, software API)?** How were these mechanisms or procedures validated?

X-ray diffraction, nuclear magnetic resonance (NMR), and electron microscopy (EM) are the most common methods for collecting new protein complexes. These techniques are industry standard in biomolecular research.

**Who was involved in the data collection process (e.g., students, crowdworkers, contractors) and how were they compensated (e.g., how much were crowdworkers paid)?**

Unknown.

**Over what timeframe was the data collected?** Does this timeframe match the creation timeframe of the data associated with the instances (e.g., recent crawl of old news articles)? If not, please describe the timeframe in which the data associated with the instances was created.

The protein structures in the RCSB PDB have been collected iteratively over the last 50 years.

**Were any ethical review processes conducted (e.g., by an institutional review board)?** If so, please provide a description of these review processes, including the outcomes, as well as a link or other access point to any supporting documentation.

Unknown.

**Does the dataset relate to people?** If not, you may skip the remaining questions in this section.

No, DIPS-Plus does not contain data that relates directly to individuals.

### A.1.4 Preprocessing, Cleaning, and Labeling

**Was any preprocessing/cleaning/labeling of the data done (e.g., discretization or bucketing, tokenization, part-of-speech tagging, SIFT feature extraction, removal of instances, processing of missing values)?** If so, please provide a description. If not, you may skip the remainder of the questions in this section.

All eight of the residue-level features added in DIPS-Plus are missing values for at least one residue. This is because not all residues have, for example, DSSP-derivable secondary structure (SS) values [39] or profile hidden Markov models (HMMs) that are derivable by HH-suite3 [35], the software package we use to generate multiple sequence alignments (MSAs) and subsequent MSA-based features. A similar situation occurs for the six other residue features. That is, not all residues have derivable features for a specific feature column, governed either by our own feature parsers or by the external feature parsers we use in making DIPS-Plus. We denote missing feature values for all features as NumPy's NaN constant with the exception of residues' SS value in which case we use '-' as the default missing feature value [43]. In the case of missing features, we substitute NumPy's NaN constant for the missing feature value. We also provide with DIPS-Plus postprocessing scripts for users to perform imputation of missing feature values (e.g. replacing a column's missing values with the column's mean, median, minimum, or maximum value or with a constant such as zero) depending on the type of the missing feature (i.e. categorical or numeric).

**Was the "raw" data saved in addition to the preprocessed/cleaned/labeled data (e.g., to support unanticipated future uses)?**

The version of each complex prior to any postprocessing we perform for DIPS-Plus complexes is saved separately in our Zenodo data repository. That is, each pruned pair from DIPS is stored in our data repository prior to the addition of DIPS-Plus features. The raw complexes from which DIPS complexes are derived can be retrieved from the RCSB PDB individually or in batch using FTP or similar file-transfer protocols (from https://ftp.wwpdb.org/pub/pdb/data/biounit/coordinates/divided/).

**Is the software used to preprocess/clean/label the instances available?** If so, please provide a link or other access point.

Our GitHub repository with source code and instructions for generating DIPS-Plus from scratch can be found at https://github.com/amorehead/DIPS-Plus.

### A.1.5 Uses

**Has the dataset been used for any tasks already?** If so, please provide a description.

At the time of publication, DIPS-Plus has been used to benchmark the performance of existing methods for PIP in Section 4 of the manuscript by training a SOTA PIP algorithm (i.e. NeiA) on DIPS-Plus and achieving SOTA results on DB5-Plus' test complexes.

**Is there a repository that links to any or all papers or systems that use the dataset?** If so, please provide a link or other access point.

We will be linking to all papers or systems that use DIPS-Plus (as we find out about them) in our GitHub repository for DIPS-Plus (https://github.com/amorehead/DIPS-Plus).

**What (other) tasks could the dataset be used for?**

This dataset can be used with most deep learning algorithms, especially geometric learning algorithms, for studying protein structures, complexes, and their inter/intra-protein interactions at scale. This dataset can also be used to test the performance of new or existing geometric learning algorithms for node classification, link prediction, object recognition, or similar benchmarking tasks.

**Is there anything about the composition of the dataset or the way it was collected and prepro-cessed/cleaned/labeled that might impact future uses?** For example, is there anything that a future user might need to know to avoid uses that could result in unfair treatment of individuals or groups (e.g., stereotyping, quality of service issues) or other undesirable harms (e.g., financial harms, legal risks) If so, please provide a description. Is there anything a future user could do to mitigate these undesirable harms?

There is minimal risk for harm: the data DIPS-Plus was created from was already public.

**Are there tasks for which the dataset should not be used?** If so, please provide a description.

This data is collected solely in the proteomics domain, so systems trained on it may or may not generalize to other tasks in the life sciences.

### A.1.6 Distribution

**Will the dataset be distributed to third parties outside of the entity (e.g., company, institution, organiza-tion) on behalf of which the dataset was created?** If so, please provide a description.

Yes, the dataset's source code is publicly available on the internet (https://github.com/amorehead/DIPS-Plus).

**How will the dataset will be distributed (e.g., tarball on website, API, GitHub)?** Does the dataset have a digital object identifier (DOI)?

The dataset is distributed on Zenodo (https://zenodo.org/record/5134732) with 10.5281/zen-odo.5134732 as its DOI.

**When will the dataset be distributed?**

The dataset has been distributed on Zenodo as of June 7th, 2021.

**Will the dataset be distributed under a copyright or other intellectual property (IP) license, and/or under applicable terms of use (ToU)?** If so, please describe this license and/or ToU, and provide a link or other access point to, or otherwise reproduce, any relevant licensing terms or ToU, as well as any fees associated with these restrictions.

The dataset will be distributed under a CC-BY 4.0 license, and the code used to generate it will be distributed on GitHub under a GPL-3.0 license. We also request that if others use the dataset they cite the corresponding paper:

*DIPS-Plus: The Enhanced Database of Interacting Protein Structures for Interface Prediction.* Alex Morehead, Chen Chen, Ada Sedova, and Jianlin Cheng. Datasets of Machine Learning Research, 2021.

**Have any third parties imposed IP-based or other restrictions on the data associated with the instances?** If so, please describe these restrictions, and provide a link or other access point to, or otherwise reproduce, any relevant licensing terms, as well as any fees associated with these restrictions.

No.

**Do any export controls or other regulatory restrictions apply to the dataset or to individual instances?** If so, please describe these restrictions, and provide a link or other access point to, or otherwise reproduce, any supporting documentation.

Unknown.

### A.1.7 Maintenance

**Who is supporting/hosting/maintaining the dataset?**

Alex Morehead (https://amorehead.github.io/) is supporting the dataset.

**How can the owner/curator/manager of the dataset be contacted (e.g., email address)?**

Alex Morehead's email address is acmwhb@missouri.edu.

**Is there an erratum?** If so, please provide a link or other access point.

No. Since DIPS-Plus was released on June 7th, 2021, there have not been any errata discovered.

**Will the dataset be updated (e.g., to correct labeling errors, add new instances, delete instances)?** If so, please describe how often, by whom, and how updates will be communicated to users (e.g., mailing list, GitHub)?

This will be posted on the dataset's GitHub repository page.

**If the dataset relates to people, are there applicable limits on the retention of the data associated with the instances (e.g., were individuals in question told that their data would be retained for a fixed period of time and then deleted)?** If so, please describe these limits and explain how they will be enforced.

N/A.

**If the dataset relates to people, are there applicable limits on the retention of the data associated with the instances (e.g., were individuals in question told that their data would be retained for a fixed period of time and then deleted)?** If so, please describe these limits and explain how they will be enforced.

If and when the dataset is updated after its initial release, we will keep older versions of it around for consistency.

**If others want to extend/augment/build on/contribute to the dataset, is there a mechanism for them to do so?** If so, please provide a description. Will these contributions be validated/verified? If so, please describe how. If not, why not? Is there a process for communicating/distributing these contributions to other users? If so, please provide a description.

Others may do so and should contact the original authors about incorporating fixes/extensions.

## A.2   Hardware and Software Used

The Oak Ridge Leadership Facility (OLCF) at the Oak Ridge National Laboratory (ORNL) is an open science computing facility that supports HPC research. The OLCF houses the Andes and Summit compute clusters. Andes is a (704)-node commodity-type Linux® cluster. Andes provides a conduit for large-scale scientific discovery via pre- and post-processing of simulation data. Each of Andes's 704 nodes contains two 16-core 3.0 GHz AMD EPYC processors and 256GB of main memory. Andes also has nine large memory GPU nodes. These nodes each have 1TB of main memory and two NVIDIA K80 GPUs with two 14-core 2.30 GHz Intel Xeon processors with HT Technology. Andes is connected to the OLCF's high-performance GPFS® filesystem, Alpine.

Summit, launched in 2018, delivers 8 times the computational performance of Titan's 18,688 nodes, using only 4,608 nodes. Like Titan, Summit has a hybrid architecture, and each node contains multiple IBM POWER9 CPUs and NVIDIA Volta GPUs all connected together with NVIDIA's high-speed NVLink. Each node has over half a terabyte of coherent memory (high bandwidth memory + DDR4) addressable by all CPUs and GPUs plus 800GB of non-volatile RAM that can be used as a burst buffer or as extended memory. To provide a high rate of I/O throughput, the nodes are connected in a non-blocking fat-tree using a dual-rail Mellanox EDR InfiniBand interconnect.

We compiled both DIPS-Plus and DB5-Plus with ORNL's Andes compute cluster, using a single compute node for inherently-sequential operations in our data postprocessing pipeline and 16 compute nodes for concurrent operations. In addition, we used Summit for our PIP method benchmarking, utilizing a single Nvidia Tesla V100 GPU (16 GB) for each of our experiments (i.e. training each model using version 1.3.8 of PyTorch Lightning [54]). We also used version 3.8.5 of Python as well as Anaconda to manage our Python dependencies. A more in-depth description of the software environment we use for constructing DIPS-Plus can be found in our GitHub repository linked above.

