# OpenReview forum: "DIPS-Plus: The Enhanced Database of Interacting Protein Structures for Interface Prediction"
_NeurIPS.cc/2021/Track/Datasets_and_Benchmarks/Round1 — Submitted to NeurIPS 2021 Datasets and Benchmarks Track (Round 1)_

### Official Review · Reviewer_ihYC · 2021-07-01
**The author just added some features to an existing dataset, and those features can be obtained from the original data**

**Rating:** 3
**Confidence:** 5

**Strengths:**

The task is meaningful in biomolecular modeling.

The methods they mentioned are machine learning methods, which are relevant to the NeurIPS community.

**Weaknesses:**

The dataset is not novel, it’s just DIPS data with some additional features. And the features are some common features that can be calculated from raw atom coordinates with some open-source tools.

There is no discussion about data quality, such as the resolution of structures from PDB.

There is no evidence that the additional features are vital.

No benchmarking of existing methods, either machine learning or domain methods.


**Additional Feedback:**

The term can be protein-protein binding site prediction or such.

**Clarity:**

The authors explained all the features they added, but didn’t explain why they chose those and if those features help.

**Correctness:**

The statement of DB5 is not correct, please see the relation to the previous work section for more detail.

The author stated the best results for interface prediction without discussing any traditional domain methods, such as global docking.


**Documentation:**

Yes, they described how they calculated the features in detail.

**Ethics:**

No.

**Relation To Prior Work:**

The authors didn’t search the field thoroughly. I understand DB5 was widely used in previous machine learning research in this field, but more datasets exist indeed. PDBBind (https://doi.org/10.1093/bioinformatics/btu626) is a PDB-wide curation of protein binding data (same approach as DIPS, just earlier) released in 2004, including 5,341 structures of protein-protein complexes up to 2015. In the interface prediction task of MaSIF (https://www.nature.com/articles/s41592-019-0666-6), they used non-duplicated data from PRISM, PDBBind, SAbDab, and ZDock, comprising 12k data in total. Thus the statement of “DIPS is 200 times larger than previous datasets” is overstated, even DIPS is not the work of authors.

**Summary And Contributions:**

The authors add some residue-level features to an existing DIPS dataset and analyze potential impact and challenges.

---

> ### Author Response · Authors · 2021-07-10
> **Response to Feedback - Point-By-Point Replies (1)**
>
> Thank you very much for your feedback on my manuscript. Below (and above) are my responses to the weaknesses you have outlined in your review (broken up into parts to meet OpenReview's character limit for each comment).
>
> $\textbf{Weakness 1:}$ "The dataset is not novel, it’s just DIPS data with some additional features. And the features are some common features that can be calculated from raw atom coordinates with some open-source tools."
>
> $\textbf{Response:}$ Respectfully, I would argue that the features generated alongside DIPS-Plus such as multiple sequence alignments (MSAs) and profile hidden Markov models (HMMs) for encoding sequence conservation features in each residue (i.e. sequence conservation information shown to be indicative of residues involved in cross-protein interactions [PAIRpred - Asfar et al. 2013; PIPGCN - Fout et al. 2017]) are valuable to the machine learning and bioinformatics community in their own right (as, to the best of my knowledge, they are not directly derivable from raw atom coordinates but most be constructed using coevolutionary information from sequence alignments). The generation of such features for the many complexes that are in DIPS-Plus is computationally prohibitive for most research groups to carry out, especially when created using large sequence databases like the Big Fantastic Database (i.e. the database we used for sequence-based features). As such, in my view, making these features available to the public is a notable research contribution as future datasets can be built off of these features using the insights gleaned from this manuscript. In addition, making our source code for DIPS-Plus open-source allows other researchers to customize or extend our data pipeline to create future datasets using complexes derived directly from the PDB. Our code allows one to add or remove filtering criteria to the complexes, generate and add any number of features to the complexes using external tools like HH-suite3 in a compute-distributed (i.e. multi-node/multiprocess) manner, construct cross-validation partitions of the complexes for model development, aggregate statistics on each complex, impute missing feature values for each complex, and convert the complexes into a graph neural network-compatible representation for deep learning. As I describe in greater detail in my response to Weakness 3 above, I have also added a list of benchmark experiments to the manuscript implying that the features we chose to use for this dataset were, in fact, novel for improving interface prediction performance using SOTA methods.

---

> ### Author Response · Authors · 2021-07-10
> **Response to Feedback - Point-By-Point Replies (2)**
>
> $\textbf{Weakness 2:}$ "There is no discussion about data quality, such as the resolution of structures from PDB."
>
> $\textbf{Response:}$ You are correct that there is little discussion about data quality in the manuscript. This was an omission originally made that has since been amended in my latest version of the manuscript. I have added a new paragraph describing the data's quality after the first paragraph in Section 3.2 (i.e. Dataset - Construction). That paragraph reads as follows:
>
> "Regarding the quality of the complexes in DIPS-Plus, we employ similar pruning methodology as [2] to ensure data integrity. DIPS-Plus, along with the works of others [1, 29, 30], derives its complexes from the PDB which conducts statistical quality summaries in its structure deposition processes and post-deposition analyses [31]. Nonetheless, recent studies on the PDB have discovered that the quality of its structures can, in some cases, vary considerably between structures [32]. As such, in selecting complexes to include in DIPS-Plus, we perform extensive filtering after obtaining the initial batch of 180,000 complexes available in the PDB. Such filtering includes (1) removing PDB complexes containing a protein chain with more than 30% sequence identity with any protein chain in DB5-Plus per [33, 34], (2) selecting complexes with an X-ray crystallography or cryo-electron microscopy resolution greater than 3.5 Å (i.e. a standard threshold in the field), (3) choosing complexes containing protein chains with more than 50 amino acids (i.e. residues), (4) electing for complexes with at least 500 $Å^{2}$ of buried surface area, and (5) picking only the first model for a given complex. The motivation for the first filtering step is to ensure that we do not allow training datasets built from DIPS-Plus to bias the DB5-Plus test results of models trained on DIPS-Plus, with the remaining steps carried out to follow conventions in the field of protein bioinformatics."
>
> In Section 3.5 (i.e. Dataset - New Features), I also mention which types of residues we excluded from DIPS-Plus:
>
> "It should be noted beforehand that these features were derived only for standard residues (e.g. amino acids) by filtering out hetero residues and waters from each PDB complex before calculating, for example, half-sphere amino acid compositions for each residue. This is, in part, to reduce the computational overhead of generating each residue's features. More importantly, however, we chose to ignore hetero residue features in DIPS-Plus to keep it consistent with DB5 as hetero residues and waters are not present in DB5."

---

> ### Author Response · Authors · 2021-07-10
> **Response to Feedback - Point-By-Point Replies (3)**
>
> $\textbf{Weakness 3:}$ "There is no evidence that the additional features are vital."
>
> $\textbf{Response:}$ To address this concern, I have added a new Section 4, Benchmarks, in my latest version of the manuscript. In it, I describe experiments run to show how a SOTA algorithm for interface prediction, NeiA, can be considerably enhanced simply by using the feature set derived in DIPS-Plus and DB5-Plus. The results we show in Section 4 (to be uploaded by 07/14/21) now reflect that we are training and validating our benchmark method (i.e. NeiA) on the entire DIPS-Plus dataset (and only the DIPS-Plus dataset) and testing on DB5-Plus's test complexes (for fair comparisons with previous methods). Our benchmarks are now designed to showcase the value proposition of training PIP methods on large-scale protein complex datasets with rich residue features (DIPS-Plus in particular). A reader with a machine learning background should now be able to reference our Benchmarks section to gain an understanding of where methods for PIP currently stand in terms of the dataset with which they trained and the standardized PIP metric (i.e. MedAUROC - a proxy for progress in the PIP literature).

---

> ### Author Response · Authors · 2021-07-10
> **Response to Feedback - Point-By-Point Replies (4)**
>
> $\textbf{Weakness 4:}$ "No benchmarking of existing methods, either machine learning or domain methods."
>
> $\textbf{Response:}$ Similar to my response to Weakness 3, my new Benchmarks section describes a set of benchmarks run with DB5 vs. DB5-Plus being used as the training and validation datasets, respectively. Regarding traditional domain methods such as global docking, I have added the following new paragraph after the second paragraph in Section 2 (i.e. Related Work):
>
> "Another avenue of research related to interface prediction stems from traditional computational approaches to protein docking. Such domain methods have previously been used to achieve global docking results between two or more protein structures, and interface predictors have found great use within such docking software. However, the performance of interface predictors remains a notable shortcoming of these traditional docking methods [DB5 - Vreven et al. 2015; Flexible Protein-Protein Docking - Bonvin et al. 2006]. Hence, innovations in interface prediction via new machine learning methods and enhanced protein complex datasets on which they are trained could lead to improved performance of future docking software."

---

> ### Author Response · Authors · 2021-07-10
> **Response to Feedback - Point-By-Point Replies (5)**
>
> $\textbf{Response to Comments in Previous Work Section:}$
> I agree that PDBBind and MaSIF's dataset need to be included as related works in the manuscript to better contextualize DIPS-Plus in the existing literature. As such, I have added a description of their relation to DIPS-Plus in Section 2 (i.e. Related Work) of the manuscript, in the first few sentences of Section 2's last paragraph:
>
> "The original DIPS, being a carefully curated PDB subset, contains almost 200x more protein complexes than the modest 230 complexes in DB5, what is still considered to be a gold-standard of protein-protein interaction datasets. Other protein binding datasets such as PDBBind[29] (containing 5,341 protein-protein complexes) and that which was used in the development of MaSIF [30] (containing roughly 12,000 protein-protein complexes in total) have previously been curated for machine learning of protein complexes. However, to the best of our knowledge, DIPS-Plus serves as the single largest database of PDB protein-protein complexes incorporating novel features such as profile HMM-derived sequence conservation and half-sphere amino acid compositions shown to be indicative of residue-residue interactions in Section 4. It is still a possibility that PDBBind or MaSIF may contain useful information regarding complexes not already contained in DIPS-Plus. Fortunately, it remains possible with our data pipeline to extend DIPS-Plus to include these new complexes in PDBBind or MaSIF. For the time being, we defer the exploration of this idea to future works."
>
> $\textbf{Follow-Up Question for the Reviewer:}$ Do you believe the above (new) excerpt better describes DIPS-Plus' place in the current literature on protein complexes?

---

> ### Author Response · Authors · 2021-07-15
> **Latest Version of Manuscript**
>
> Between 7/7/21 and now, many revisions have been made to the manuscript linked above (https://openreview.net/pdf?id=lwlkxYsGDi). We recommend referencing its latest version going forward. Thank you once again for your time and feedback.

---

### Official Review · Reviewer_rfBk · 2021-07-02
**Review of DIPS-Plus: The Enhanced Database of Interacting Protein Structures for Interface Prediction**

**Rating:** 4
**Confidence:** 4

**Strengths:**

As a computational biologist, I can definitely appreciate the utility of this dataset, and the challenges in producing it. Calculating these features for so many complexes involves a great deal of computational resources - the authors efforts in making this a public dataset will make these features accessible to many groups that would not otherwise have the time or hardware. I also think that the scale of this dataset goes very well in hand with current machine learning research for proteins: while there has been some success in unsupervised representation learning research directly from sequence using large-scale protein databases like Uniprot or Pfam, to my knowledge, large-scale models have not been trained on residue-level features, which I suspect is due to the lack of accessibility of large enough datasets of this nature. Hence, this dataset could potentially spur new methods.

**Weaknesses:**

My main issue with the paper is that it is currently not written in a way that a general machine learning audience will appreciate the dataset. The summary of the problem is okay, but the authors do not review prior literature sufficiently to establish an argument for why this dataset is useful, and how they anticipate it will be used. For a non-biologist, it is not clear what the difference between atom-level or residue-level features is. Even for a more specialist computational biologist like myself, I'm not entirely clear on what kind of difference the authors expect providing residue-level features, as opposed to atom-level features or just direct sequence, will make for current machine learning methods - is this simply an argument from principle, or is there any evidence that these features truly provide enough additional features for models to "generalize" well? This seems to be a really major premise of the paper - I would like the authors to better review the actual methods that have been previously built on DB5-Plus to convince the reader that residue-level features are a big deal.

Additionally, the authors have not done enough to contextualize their work in respect to current feature learning approaches on proteins. The authors touch upon how deep learning has established itself as an effective means of automatically learning features from data - but they seem to have completely missed the major representation learning advances using deep learning to learn a mapping from raw sequence to feature representations. Many of these previous works have shown that these learned features correlate with many of the more classic bioinformatics features that the authors have described here like secondary structure. I'd like a better discussion of the trade-offs between classic features that the authors provide here versus the learned features: is there any reason to believe that these classic features are more conductive to performance than recent unsupervised protein features, or are the advantages more on other axes like interpretability? I know in many bioinformatics disciplines, e.g. especially bioimaging, even unsupervised or self-supervised deep learning representations have blown away expert-designed knowledge-based features on classification and regression tasks.

Finally, there are weaknesses from a structural biology perspective (see feedback). The authors spend a lot of time delving deep into the features they use, which are rather routine, so the paper reads like a manual. My impression is that the key challenge in this dataset is not the feature production, but the quality of the PDB structures given the sheer volume of data, not all of it reliable. From what I can tell, there has not been quality checks on the data on the level of basic aspects like clashes or filters for resolution (or at least the authors do not explain this), so I am uncertain about the overall quality of the data as a whole.

**Additional Feedback:**

1. The authors go into depth on explaining their features, but they spend no time addressing the known errors, biases, and artifacts in the structures themselves, which will inevitably influence the quality of the computed features. Did the authors perform any quality checks on these structures?  i.e. What is the resolution limit of the structure of a complex for entry in their dataset? What side-chain resolution do they tolerate? Are positions of side-chains in the structures checked for clashes, and if those are encountered, fixed by e.g. a molecular dynamics protocol? There is a lot more I could state, but including even just these basic things would help.

2. How exactly do their features address the physics of the interactions between main-chain and side-chain groups? Physicochemical features of different amino-acids should be most informative for learning -- those I guess these are implicitly embedded in features such as protrusion index and residue depth - but more discussion on this would nonetheless be useful.

**Clarity:**

The language itself is fine, but the paper is not written with sufficient context, motivation, or background for a general machine learning audience to understand the contributions of the paper.

**Correctness:**

Yes - the features calculated are standard bioinformatics features, and I see no major issues with their definition.

**Documentation:**

Yes: the dataset has been uploaded to a stable public repository.

**Ethics:**

No: in its current iteration, the dataset is more intended to support basic bioinformatics research, and there are not many major risks or ethical issues associated with this research.

**Relation To Prior Work:**

No, this is not sufficient: see weaknesses.

**Summary And Contributions:**

Here, the authors provide a dataset of features, calculated on a per-residue level, for 42,112 protein complexes. This dataset is intended to contribute towards the problem of interface prediction between putative protein interaction partners. While previous datasets of atom-level features exist on a similar scale, their dataset is currently the largest for classic engineered residue-level features, improving the scale of these datasets by two orders of magnitude.

---

> ### Author Response · Authors · 2021-07-10
> **Response to Feedback - Point-By-Point Replies (1)**
>
> Thank you very much for your detailed feedback on my manuscript. I am especially grateful for the level of detail with which you examined its contents. Below (and above) are my responses to the weaknesses you have outlined in your review (broken up into parts to meet OpenReview's character limit for each comment).
>
> $\textbf{Weakness 1:}$ "My main issue with the paper is that it is currently not written in a way that a general machine learning audience will appreciate the dataset."
>
> $\textbf{Response:}$ To better contextualize DIPS-Plus and similar protein complex databases, I have added the following description of traditional domain methods such as global docking and their need for interface predictors after the second paragraph in Section 2 (i.e. Related Work):
>
> "Another avenue of research related to interface prediction stems from traditional computational approaches to protein docking. Such domain methods have previously been used to achieve global docking results between two or more protein structures, and interface predictors have found great use within such docking software. However, the performance of interface predictors remains a notable shortcoming of these traditional docking methods [DB5 - Vreven et al. 2015; Flexible Protein-Protein Docking - Bonvin et al. 2006]. Hence, innovations in interface prediction via new machine learning methods and enhanced protein complex datasets on which they are trained could lead to improved performance of future docking software."
>
> I would also like to point to the data sheet listed in the manuscript's supplementary material, specifically where it contains a section titled $\textit{What (other) tasks could the dataset be used for?}$. This write-up expounds on DIPS-Plus' possible use cases for a general machine learning audience (i.e. on page 16 of the manuscript's latest version):
>
> "This dataset can be used with most deep learning algorithms, especially geometric learning algorithms, for studying protein structures, complexes, and their inter/intra-protein interactions at scale. This dataset can also be used to test the performance of new or existing geometric learning algorithms for node classification, link prediction, object recognition, or similar benchmarking tasks."
>
> I have since integrated these statements into the body of the manuscript, specifically into the first three sentences of the first paragraph in Section 3.2 (i.e. Dataset - Construction).

---

> ### Author Response · Authors · 2021-07-10
> **Response to Feedback - Point-By-Point Replies (2)**
>
> $\textbf{Weakness 2:}$ "For a non-biologist, it is not clear what the difference between atom-level or residue-level features is."
>
> $\textbf{Response:}$ I agree that the difference between these two protein representation schemes was not communicated in a clear way for a general machine learning audience. To remedy this potential source of confusion, I have made an amendment to the first paragraph of Section 3.1 (i.e. Dataset - Overview) such that the first half of the paragraph now reads as follows:
>
> "As we have seen, two main encoding schemes have been proposed for protein interface prediction: modeling protein structures at the atomic level and modeling structures at the level of the residue. Modeling protein structures in terms of their individual atoms can yield a detailed representation of such geometries, however, accounting for each atom in a structure can quickly become computationally burdensome or infeasible for large structures. On the other hand, as residues are comprised of multiple atoms, modeling only a structure's residues allows one to employ their models on a more computationally succinct view of the structure, thereby reducing memory requirements for the training and inference of biomolecular machine learning models and allowing one to swiftly experiment with more advanced architectures. The latter scheme also enables researchers to curate robust residue-based features for a particular task, a notion of flexibility quite important to the success of prior works in protein bioinformatics [Asfar et al. 2014, Townshend et al. 2019, Liu et al. 2020, Guo et al. 2021]. Nonetheless, both schemes, when adopted by a machine learning algorithm such as a neural network, require copious amounts of training examples to generalize past the training dataset. However, only a handful of extensive datasets for protein interface prediction currently exist, DIPS being the largest of such examples, and it is designed solely for modeling structures at the atomic level. If one would like to model complexes at the residue level to summarize the structural and functional properties of each residue's atoms as additional features for training, DB5 is currently one of the only datasets with readily-available pairwise residue labels that meets this criterion."
>
> In addition to this new description of the encoding schemes, to address the importance of a residue-level encoding of proteins, I have added a new Section 4, Benchmarks, in my latest version of the manuscript. In it, I describe experiments run to show how a SOTA algorithm for interface prediction, NeiA, can be considerably enhanced simply by using the feature set derived in DIPS-Plus and DB5-Plus, suggesting that our choice of residue-level features are very useful for machine learning models to be able to achieve state-of-the-art performance in distinguishing between interacting and non-interacting residue pairs. I also contextualize the results by comparing them to the best PIP performance of other machine learning models trained on only the atom or residue-level features of protein complexes for interface prediction (of which I am aware) to show that our residue-level protein encoding yields significant improvements for the task.

---

> ### Author Response · Authors · 2021-07-10
> **Response to Feedback - Point-By-Point Replies (3)**
>
> $\textbf{Weakness 3:}$ "Additionally, the authors have not done enough to contextualize their work in respect to current feature learning approaches on proteins."
>
> $\textbf{Response:}$ I agree that the manuscript originally did not provide enough background to readers about recent advances in unsupervised learning of protein features. I have since amended it to include a discussion of the trade-offs between using classical features and using features learned via unsupervised learning on protein sequences (e.g. MSA Transformer), adding the discussion directly following a new related works paragraph on traditional domain methods (making the following paragraph now the new fourth paragraph in Section 2):
>
> "Over the past several years, deep learning has established itself as an effective means of automatically learning useful feature representations from data, with the MSA Transformer presenting a prime example of successful unsupervised learning on protein sequences [Rao et al. 2021]. Rivaling classical features, these learned feature representations, which oftentimes describe complex interactions and relationships between entities, can be used for a range of tasks including classification, regression, generative modeling, and even advanced tasks such as playing Go [Silver et al. 2016] or folding proteins $\textit{in silico}$ [Jumper et al. 2020]. On the other hand, these learned feature representations come at the cost of interpretability and task specificity. For example, in interface prediction, knowledge of amino acids' side-chain geometry in $\mathbb{R}^{3}$ presents crucial structural information that would otherwise have to be learned $\textit{a priori}$ by an unsupervised learning algorithm and would be more difficult to describe the contributions of to the network's prediction."

---

> ### Author Response · Authors · 2021-07-10
> **Response to Feedback - Point-By-Point Replies (4)**
>
> $\textbf{Weakness 4:}$ "Finally, there are weaknesses from a structural biology perspective (see feedback)."
>
> $\textbf{Response:}$ I agree that there was little discussion about data quality in the original manuscript. This was an omission that has since been amended in my latest version of the manuscript. I have added a new paragraph describing the data's quality after the first paragraph in Section 3.2 (i.e. Dataset - Construction). That paragraph reads as follows:
>
> "Regarding the quality of the complexes in DIPS-Plus, we employ similar pruning methodology as [Townshend et al. 2019] to ensure data integrity. DIPS-Plus, along with the works of others [DB5 - Vreven et al. 2015; Liu et al. 2015; Gainza et al. 2020], derives its complexes from the PDB which employs rigorous quality checking in its structure deposition processes and post-deposition analyses [Worldwide Protein Data Bank validation information - Smart et al. 2018]. As such, the filtering we perform after obtaining the initial batch of 160,000 complexes available in the PDB is founded on a data source which, for good reason, has historically been trusted by the bioinformatics community. Such filtering includes (1) removing complexes containing proteins with more than 30% sequence identity with any protein in DB5-Plus per [Jordan et al. 2012; Yang et al. 2013], (2) selecting complexes derived with an X-ray crystallography or cryo-electron microscopy resolution greater than 3.5 $Angstrom$ (i.e. a standard threshold in the field), (3) choosing complexes containing protein chains with more than 50 amino acids (i.e. residues), (4) electing for complexes with at least 500 $Angstrom^{2}$ of buried surface area, and (5) picking only the first model for a given complex. The motivation for the first filtering step is to ensure that we do not allow training datasets built from DIPS-Plus to bias the DB5-Plus test results of models trained on DIPS-Plus, with the remaining steps carried out to follow conventions in the field of protein bioinformatics."

---

> ### Author Response · Authors · 2021-07-10
> **Response to Feedback - Point-By-Point Replies (5)**
>
> $\textbf{Weakness 5:}$ "The authors go into depth on explaining their features, but they spend no time addressing the known errors, biases, and artifacts in the structures themselves, which will inevitably influence the quality of the computed features."
>
> $\textbf{Response:}$ I believe my response to Weakness 4 below addresses many of these concerns, however, if any questions or uncertainties about the dataset's quality remain, please do not hesitate to make them known.

---

> ### Author Response · Authors · 2021-07-10
> **Response to Feedback - Point-By-Point Replies (6)**
>
> $\textbf{Weakness 6:}$ "How exactly do their features address the physics of the interactions between main-chain and side-chain groups?"
>
> $\textbf{Response:}$ The secondary structures of residues have been found to be informative of these physical interactions between main-chain and side-chain groups [Main-chain conformational features at different conformations of the side-chains in proteins - Chakrabarti et al. 1998]. That was one of the original motivations for including them as a residue feature in DIPS-Plus. I have made this point more explicit in the manuscript by amending the first half of Section 3.3.1 (i.e. Dataset - New Features - Secondary Structure) to read as follows:
>
> "Secondary structure (SS) is included in DIPS-Plus as a categorical variable that describes the type of local, three-dimensional structural segment in which a residue can be found. This feature has been shown to correlate with the presence or absence of protein-protein interfaces [Taechalertpaisarn et al. 2019]. In addition, the secondary structures of residues have been found to be informative of the physical interactions between main-chain and side-chain groups [Chakrabarti et al. 1998]. This is one of the primary motivations for including them as a residue feature in DIPS-Plus. As such, we hypothesize adding secondary structure as a feature for interface prediction models could prove beneficial to model performance as it would allow them to more readily discover interactions between structure's main-chain and side-chain groups."
>
> Regarding physiochemical features of amino acids, as you suggested, these are implicitly described by a residue's half-sphere amino acid composition (HSAAC). HSAACs detail physiochemical as well as geometric patterns in the region facing and opposite to a residue's side-chain [An amino acid has two sides - Hamelryck et al. 2005]. I have added some more detail regarding this contribution by HSAACs to the last half of Section 3.3.5 (i.e. Dataset - New Features - Half-Sphere Amino Acid Composition):
>
> "Knowing the composition of amino acids along and away from a residue's side chain, for all residues in a structure, is another feature that has been shown to offer crucial predictive value to machine learning models for interface prediction as it can describe physiochemical and geometric patterns in such regions [Hamelryck et al. 2005]. These UC and DC vectors can also vary widely for residues, suggesting an alternative way of assessing residue accessibility [Afsar et al. 2014, Liu et al. 2020]."

---

> > ### Comment · Reviewer_rfBk · 2021-07-12
> > **Response to replies**
> >
> > Thank you to the authors for their work in improving the manuscript: I think they've added some more context, and their addition of a benchmark supports some of their arguments. However, I find that the overall manuscript still lacks clarity for a general audience, and I'm not finding the premise of the dataset to be convincing for this audience, at least in the way it's framed.
> >
> > 1. While the authors have somewhat improved this issue on several fronts, I still find the paper inaccessible and unconvincing for a general machine learning audience. While computational biologists, or people who are already working on this problem might find a way to make this dataset useful, a general audience with limited exposure to protein interface prediction will not. The paper currently reads like it is providing an unstructured dump of data: there are limited attempts to provide a standardized task or metrics for understanding progress on the problem, there is no standardized test/train split, and even their own benchmarks treat the dataset as a pre-training corpus for another dataset. Unstructured data can be useful (see the original ImageNet before it was reorganized into a classification dataset or Wikipedia), but the problem here is that the dataset only reflects one possible approach to solving the problem (this is the reason why I emphasized self-supervised feature representations in my original review), which the authors further pigeonhole by emphasizing GNNs above other possible approaches. There's no real convincing case the authors make for why practitioners should solve this problem in the one limited way that their dataset is good for.
> >
> > 2. The new benchmarks contain many details that raise confusion for me, or otherwise contradict the claims of the paper. Why do the authors use the DB5-Plus test complexes, instead of evaluating on a test split of DIPS-Plus? A previous claim was that DB5-Plus was too small: is this test dataset really indicative of predictiveness on protein interface prediction in general, given it's only 55 complexes? The authors also make a big deal about the quality of DIPS-Plus - why is it not appropriate to just use a test split of DIPS-Plus directly? (Note that this does not mean I think an automatically filtered dataset is equivalent to a manually curated one - the point is that the limitations of DIP-Plus are not discussed, so this will be lost a general machine learning audience.) Why do the authors only use 2% of DIPS-Plus for pre-training? This puts the dataset at the same order of magnitude of the previous ones - the authors made a big deal previously about the large scale of the dataset as being important. Moreover, it makes the benchmarks less useful for future work, because there are still no benchmarks that actually use the full dataset. And why is the fine-tuning procedure on DB5-Plus necessary? Overall, while the benchmarks do establish that residue-level features can be useful over atom-level features, they do not provide an unambiguous problem setting, or provide a useful baseline for future work with this dataset. I would have no problem with these benchmarks if they were provided as a paper whose primary point was "residue-level features are additionally useful over atomic-level features", but this is a dataset paper which makes the claim that the dataset itself is useful to researchers beyond the authors - these benchmarks do not really contribute to that point.
> >
> > 3. I agree with reviewer kNff's comments about the unscientific language in the paper: the language is editorialized. My issue is not that the language is arbitrarily distasteful, but rather that it leads to the paper reading as making claims that are subjective, or not supported by literature or scientific consensus. These issues are still present in the current version of the manuscript. As just a few examples: Line 67 "presents crucial structural information that would otherwise have to be fortuitously learned" - "fortuitiously" implies that these methods are learning these features by chance, but the feature learning is actually dictated by the proxy task, which does have some principles associated in what features will be learned. Line 146 "historically been trusted by the bioinformatics community" - I agree that PDB is generally more reliable than many other bioinformatics repositories, but there still has been a lot of discussion about the inconsistent quality of deposits (e.g. see Domagalski et al. 2014).  Line 156 "the frightfully-small 230 complexes in DB5" - "frightful" seems like a very subjective phrase here. There are other instances, so I would overall encourage the authors to be more precise about their language, and consider what claims they're willing to support.

---

> > > ### Comment · Reviewer_rfBk · 2021-07-13
> > > **A few other comments...**
> > >
> > > I'm also including a few other comments from a structural biology perspective, in case it helps the authors make improvements to the paper:
> > >
> > > 1. It's not clear how the authors are curating data from PDB for this benchmark. A quick look at the PDB statistics tell us at 30% sequence identity, there are a total of 41,757 entries for unique protein chains (note this is not complexes). However, the authors get to 42,112 protein complexes at 30% sequence identity, after all of their filters, so something is not matching up. For this work to be reproducible, the authors need to provide their query search term for the PDB, or otherwise explain their process. Another statement of concern is that the authors claim that the initial batch of complexes they started from was ~160,000 - but the PDB has altogether ~180,000 entries, the vast majority being monomers <60 kDA. Something is going wrong with the numbers, or else the authors are not defining complexes in a commonly understood way.
> > >
> > > 2. At the resolution of 3.5 Angstroms, one cannot often confidently position side-chains in the electron density. Some refinement on the side-chains of the structures with resolution above 2.5 Angstroms is necessary if one is to extract any specific side-chain-main/side chain interaction details at the interface. With this filter alone, the quality of the extracted features remains questionable.
> > >
> > > 3. I alluded to this before, but the statement that "the PDB employs rigorous quality checking in its structure deposition processes and post-deposition analyses" is somewhat misleading. It is true that the PDB does quality checks, but this is only to give statistics on the quality of the structures, not to prevent people from depositing poor quality data. It is contingent on the end-user to apply stringent criteria for what will form their dataset based on their research questions. So while some basic filters have been performed, my overall assessment is that they are not rigorous enough (see point 2) to really ensure that the data is of high-quality.

---

> > > > ### Author Response · Authors · 2021-07-14
> > > > **Addressing Latest Replies (4)**
> > > >
> > > > $\textbf{Weakness 6:}$ "It's not clear how the authors are curating data from PDB for this benchmark. A quick look at the PDB statistics tell us at 30\% sequence identity, there are a total of 41,757 entries for unique protein chains (note this is not complexes). However, the authors get to 42,112 protein complexes at 30\% sequence identity, after all of their filters, so something is not matching up. For this work to be reproducible, the authors need to provide their query search term for the PDB, or otherwise explain their process. Another statement of concern is that the authors claim that the initial batch of complexes they started from was ~160,000 - but the PDB has altogether ~180,000 entries, the vast majority being monomers <60 kDA. Something is going wrong with the numbers, or else the authors are not defining complexes in a commonly understood way."
> > > >
> > > > $\textbf{Response:}$ To clarify, the 30\% sequence identity is in regards to preventing cross-contamination between protein chains in DIPS-Plus and DB5-Plus. The motivation for this filtering step is to ensure that models trained on DIPS-Plus are not overly familiar with any protein chain in DB5-Plus during testing. Applying this filtering criterion, followed by our four other filtering steps, yields 42,112 complexes (as of April 2021, when we last downloaded all PDB complexes to build DIPS-Plus). In essence, we are not performing sequence identity comparisons between each PDB complex's chains but rather between a prospective PDB complex (i.e. its chains) and all chains in DB5-Plus.
> > > >
> > > > (From the (New) Section 3.4 - Dataset -> Quality)
> > > >
> > > > "Such filtering includes (1) removing PDB complexes containing a protein chain with more than 30\% sequence identity with any protein chain in DB5-Plus per [Jordan et al. 2012; Yang et al. 2013]"
> > > >
> > > > Regarding how we query for complexes from the PDB, this information is provided in much more detail in our GitHub repository accompanying this manuscript (https://github.com/amorehead/DIPS-Plus). In short, we use the File Transfer Protocol (FTP) to fetch each available complex from the PDB using the following rsync command and apply various levels of post-processing to the received complexes thereafter:
> > > >
> > > > rsync -rlpt -v -z --delete --port=33444 --include='*.gz' --include='*.xz' --include='*/' --exclude '*' \
> > > > rsync.rcsb.org::ftp\_data/biounit/coordinates/divided/ project/datasets/DIPS/raw/pdb
> > > >
> > > > You are correct that the PDB currently contains 180,000 total complexes, not 160,000 as I had mentioned before. I have since amended my wording in the manuscript. Regarding how we define complexes, for each complex downloaded from the PDB via FTP, we look at each of its chains to find what we term "neighboring" chains (i.e. those with at least one non-heavy atom within 6 Angstrom of a non-heavy atom in the opposite chain). Using this definition of a complex, we can obtain from a PDB complex $\textit{multiple}$ binary (i.e. two-protein) complexes for training and validation. This accumulation of "neighboring chains" in each PDB complex occurs prior to any of the in-depth filtering criteria described in our discussions previously as well as in Section 3.4 (Dataset -> Quality) of the manuscript (e.g. sequence identity threshold, resolution requirement).

---

> > > > ### Author Response · Authors · 2021-07-14
> > > > **Addressing Latest Replies (5)**
> > > >
> > > > $\textbf{Weakness 7:}$ "At the resolution of 3.5 Angstroms, one cannot often confidently position side-chains in the electron density. Some refinement on the side-chains of the structures with resolution above 2.5 Angstroms is necessary if one is to extract any specific side-chain-main/side chain interaction details at the interface. With this filter alone, the quality of the extracted features remains questionable."
> > > >
> > > > $\textbf{Response:}$ The threshold we use, 3.5 Angstrom, was decided upon by reviewing previous works that make use of PDB structures for quality-critical tasks. Based on our review [e.g. Implication for alphavirus host-cell entry - Chen et al. 2018; End-to-End Learning - Townshend et al. 2019; Cryo-EM map interpretation and protein model-building - Terwilliger et al. 2019], we concluded that 3.5 Angstrom was a reasonable threshold to choose to strike a balance between quality and quantity of protein complexes available to users. As always, since we have open-sourced the code to our GitHub repository for DIPS-Plus, others are able and welcome to modify our data generation, filtering, and processing pipeline to accommodate a 2.5 Angstrom resolution (if they so choose).
> > > >
> > > > $\textbf{Weakness 8:}$ "I alluded to this before, but the statement that "the PDB employs rigorous quality checking in its structure deposition processes and post-deposition analyses" is somewhat misleading. It is true that the PDB does quality checks, but this is only to give statistics on the quality of the structures, not to prevent people from depositing poor quality data. It is contingent on the end-user to apply stringent criteria for what will form their dataset based on their research questions. So while some basic filters have been performed, my overall assessment is that they are not rigorous enough (see point 2) to really ensure that the data is of high-quality."
> > > >
> > > > $\textbf{Response:}$ I have made amendments in the manuscript to loosen our claims on the quality of initial PDB complexes. Regarding the quality of DIPS-Plus' complexes, the filtering criteria we employ to select them follow previous works in the protein complex literature [e.g. Jordan et al. 2012; Yang et al. 2013; Townshend et al. 2019]. In addition, our benchmark results (to be uploaded 07/14/21) support our claim that DIPS-Plus' complexes are of desirable quality for interface prediction in that training models on DIPS-Plus can lead to state-of-the-art results for PIP.

---

> > > ### Author Response · Authors · 2021-07-14
> > > **Addressing Latest Replies (1)**
> > >
> > > Thank you once again for your detailed feedback on my latest version of the manuscript.
> > >
> > > $\textbf{Weakness 1:}$ "While the authors have somewhat improved this issue on several fronts, I still find the paper inaccessible and unconvincing for a general machine learning audience. While computational biologists, or people who are already working on this problem might find a way to make this dataset useful, a general audience with limited exposure to protein interface prediction will not. The paper currently reads like it is providing an unstructured dump of data: there are limited attempts to provide a standardized task or metrics for understanding progress on the problem, there is no standardized test/train split, and even their own benchmarks treat the dataset as a pre-training corpus for another dataset."
> > >
> > > $\textbf{Response:}$ Some of these key details were indeed missing in previous versions of the manuscript. I have since added Section 3.2 (Dataset - Usage) to start by emphasizing how this dataset can be adopted by a general machine learning audience, specifically by providing details about the dataset's standardized metric, task, and cross-validation splits for machine learning researchers and practitioners to use.
> > >
> > > $\textbf{Weakness 2:}$ "Unstructured data can be useful (see the original ImageNet before it was reorganized into a classification dataset or Wikipedia), but the problem here is that the dataset only reflects one possible approach to solving the problem (this is the reason why I emphasized self-supervised feature representations in my original review), which the authors further pigeonhole by emphasizing GNNs above other possible approaches. There's no real convincing case the authors make for why practitioners should solve this problem in the one limited way that their dataset is good for."
> > >
> > > $\textbf{Response:}$ Based on this feedback, we have made efforts in the latest version of the manuscript to standardize how machine learning researchers and practitioners could make use of DIPS-Plus when developing their methods, regardless of whether they choose to use CNNs, GNNs, or other geometric deep learning algorithms on DIPS-Plus. We emphasize this last point in our new Section 3.2 (Dataset - Usage) and also in our updated Benchmarks section by showcasing the benchmark results of previous geometric deep learning methods besides GNNs. We have also edited the language throughout the manuscript to reflect the broad applicability of DIPS-Plus in PIP method development and evaluation. To concretize this point, we note for readers that all processed data in DIPS-Plus can be extracted, without any loss of information, between GNNs and other forms of networks as each complex is represented by a collection of generic PyTorch tensors.

---

> > > ### Author Response · Authors · 2021-07-14
> > > **Addressing Latest Replies (2)**
> > >
> > > $\textbf{Weakness 3:}$ "The new benchmarks contain many details that raise confusion for me, or otherwise contradict the claims of the paper. Why do the authors use the DB5-Plus test complexes, instead of evaluating on a test split of DIPS-Plus? A previous claim was that DB5-Plus was too small: is this test dataset really indicative of predictiveness on protein interface prediction in general, given it's only 55 complexes? The authors also make a big deal about the quality of DIPS-Plus - why is it not appropriate to just use a test split of DIPS-Plus directly?"
> > >
> > > $\textbf{Response:}$ To clarify the rationale behind our choice of test dataset, we use DB5-Plus' $\textit{unbound}$ test complexes to evaluate our benchmarks for the following reasons:
> > >
> > > (From the (New) Section 3.2 - Dataset -> Usage)
> > >
> > > "(1) The task of interface prediction is to predict how two \textit{unbound} (i.e. not necessarily conformal) proteins will bind together by predicting which pairs of residues from each complex will interact with one another upon binding; (2) DIPS-Plus consists solely of $\textit{bound}$ protein complexes (i.e. those already conformed to one another), so we must test on a dataset consisting of $\textit{unbound}$ complexes after training to verify the effectiveness of the method for PIP; (3) Each of DB5-Plus' $\textit{unbound}$ test complexes are of varying difficulties and interaction types for prediction (e.g. antibody-antigen, enzyme substrate), simulating how future unseen proteins (i.e. those in the wild) might be presented to the model following its deployment; (4) DB5's test complexes (i.e. those added between DB4 and DB5) represent a time-based data split also used for evaluation in [Fout et al. 2017, Townshend et al. 2019, Liu et al. 2020], so for fair comparison with previous SOTA methods we chose the same complexes for testing."
> > >
> > > $\textbf{Weakness 4:}$ "(Note that this does not mean I think an automatically filtered dataset is equivalent to a manually curated one - the point is that the limitations of DIP-Plus are not discussed, so this will be lost a general machine learning audience.) Why do the authors only use 2\% of DIPS-Plus for pre-training? This puts the dataset at the same order of magnitude of the previous ones - the authors made a big deal previously about the large scale of the dataset as being important. Moreover, it makes the benchmarks less useful for future work, because there are still no benchmarks that actually use the full dataset. And why is the fine-tuning procedure on DB5-Plus necessary? Overall, while the benchmarks do establish that residue-level features can be useful over atom-level features, they do not provide an unambiguous problem setting, or provide a useful baseline for future work with this dataset. I would have no problem with these benchmarks if they were provided as a paper whose primary point was "residue-level features are additionally useful over atomic-level features", but this is a dataset paper which makes the claim that the dataset itself is useful to researchers beyond the authors - these benchmarks do not really contribute to that point."
> > >
> > > $\textbf{Response:}$ The original motivation for pre-training on 2\% of DIPS-Plus was to make use of all available (non-redundant) data in DB5-Plus during fine-tuning and establish an upper bound on the time it takes to train each of our models. However, based on your latest feedback, we have since decided to forgo fine-tuning on DB5-Plus. The results we show in Section 4 (to be uploaded by 07/14/21) now reflect that we are training and validating our benchmark method (i.e. NeiA) on the entire DIPS-Plus dataset (and only the DIPS-Plus dataset) and testing on DB5-Plus's test complexes (for fair comparisons with previous methods). We have included the results from these latest experiments in Section 4 of the manuscript (Benchmarks). Our benchmarks are now designed to showcase the value proposition of training PIP methods on large-scale protein complex datasets with rich residue features (DIPS-Plus in particular). A reader with a machine learning background should now be able to reference our Benchmarks section to gain an understanding of where methods for PIP currently stand in terms of the dataset with which they trained and the standardized PIP metric (i.e. MedAUROC - a proxy for progress in the PIP literature).

---

> > > ### Author Response · Authors · 2021-07-14
> > > **Addressing Latest Replies (3)**
> > >
> > > $\textbf{Weakness 5:}$ "I agree with reviewer kNff's comments about the unscientific language in the paper: the language is editorialized. My issue is not that the language is arbitrarily distasteful, but rather that it leads to the paper reading as making claims that are subjective, or not supported by literature or scientific consensus. These issues are still present in the current version of the manuscript. As just a few examples: Line 67 "presents crucial structural information that would otherwise have to be fortuitously learned" - "fortuitiously" implies that these methods are learning these features by chance, but the feature learning is actually dictated by the proxy task, which does have some principles associated in what features will be learned. Line 146 "historically been trusted by the bioinformatics community" - I agree that PDB is generally more reliable than many other bioinformatics repositories, but there still has been a lot of discussion about the inconsistent quality of deposits (e.g. see Domagalski et al. 2014). Line 156 "the frightfully-small 230 complexes in DB5" - "frightful" seems like a very subjective phrase here. There are other instances, so I would overall encourage the authors to be more precise about their language, and consider what claims they're willing to support."
> > >
> > > $\textbf{Response:}$ I agree that "fortuitously" on Line 67 implies the wrong connotation. I have since removed it along with other uses of editorial language throughout the manuscript. I have also added a bit more discussion regarding the possible limitations of the PDB and our approach to navigating them when constructing DIPS-Plus, attempting to keep the language more well-grounded in the literature:
> > >
> > > (From the (New) Section 3.4 - Dataset -> Quality)
> > >
> > > "DIPS-Plus, along with the works of others [Vreven et al. 2015; Liu et al. 2015; Gainza et al. 2020], derives its complexes from the PDB which conducts statistical quality summaries in its structure deposition processes and post-deposition analyses [Smart et al. 2018]. Nonetheless, recent studies on the PDB have discovered that the quality of its structures can, in some cases, still vary considerably between structures [Domagalski et al. 2014]. As such, in selecting complexes to include in DIPS-Plus, we perform extensive filtering after obtaining the initial batch of 180,000 complexes available in the PDB."

---

> ### Author Response · Authors · 2021-07-15
> **Latest Version of Manuscript**
>
> Between 7/7/21 and now, many revisions have been made to the manuscript linked above (https://openreview.net/pdf?id=lwlkxYsGDi). We recommend referencing its latest version going forward. Thank you once again for your time and feedback.

---

### Official Review · Reviewer_kNff · 2021-07-04
**Clear contribution for facilitating prediction of protein-protein interface based on the DIPS dataset**

**Rating:** 7
**Confidence:** 1
**Clarity:** The paper is clear and well written.

**Strengths:**

**Significance:** understanding protein-protein interface regions could aid in discovering new drugs and engineered proteins

**Relevance:** this work is relevant to those working at the intersection of ML and biology

**Accessibility:** this work makes the existing DIPS dataset more accessible to ML researchers

**Weaknesses:**

I do not know enough about the area to assess the weaknesses.

**Additional Feedback:**

The authors use unprofessional negative language such as "paltry" to refer to the DB5 dataset. This seems unnecessary and should be removed. That language is not needed to emphasize the benefits of the DIPS dataset over the DB5 dataset.

**Correctness:**

I do not know enough about the area to assess the correctness, though the work is presented clearly.

**Documentation:**

The documentation in both the paper and on the GitHub appear clear, though I have not tried to access the dataset directly. There is a clear maintenance plan described in the paper.

**Ethics:**

Not that I can think of.

**Relation To Prior Work:**

Yes, the authors discuss both a previous dataset (DB5,DB5-PLUS) along with the original DIPS dataset in detail.

**Summary And Contributions:**

The authors present the DIPS-PLUS dataset, which builds on the DIPS dataset to enable prediction of protein-protein interfaces.

---

> ### Author Response · Authors · 2021-07-10
> **Response to Feedback - Change of Wording in Abstract**
>
> Thank you very much for your feedback on my manuscript. I have amended my poor choice of an adjective in the abstract (i.e. "paltry" to "modest") for the latest version of the manuscript.

---

> ### Author Response · Authors · 2021-07-15
> **Latest Version of Manuscript**
>
> Between 7/7/21 and now, many revisions have been made to the manuscript linked above (https://openreview.net/pdf?id=lwlkxYsGDi). We recommend referencing its latest version going forward. Thank you once again for your time and feedback.

---

### Decision · Program_Chairs · 2021-07-27

**Decision:**

Reject

**Comment:**

The authors present a dataset of features for 42,112 protein complexes that augment the DIPS dataset. The reviewers noted importance of the task and the possible applications for this dataset, but raise significant concerns about the novelty and the clarity of this work. While they have noted the improved new version submitted by authors, the reviewers still recommend against accepting the paper after the authors' response.